# The Evolution of Patient Satisfaction in Postoperative Care: The Impact of Investments and the Algorithm for Assessing Significant Growth over the Last 5 Years

**DOI:** 10.3390/healthcare12181824

**Published:** 2024-09-12

**Authors:** Adriana Vladu, Timea Claudia Ghitea, Lucia Georgeta Daina, Codrin Dan Nicolae Ilea, Mădălina Diana Daina

**Affiliations:** 1Faculty of Medicine and Pharmacy, Doctoral School, University of Oradea, 1 December Sq., 410081 Oradea, Romania; adrianavladu68@yahoo.com (A.V.); codrin.ilea@csud.uoradea.ro (C.D.N.I.); diana_daina98@yahoo.com (M.D.D.); 2Pharmacy Department, Faculty of Medicine and Pharmacy, University of Oradea, 1 December Sq., 410081 Oradea, Romania; 3Psycho-Neurosciences and Recovery Department, Faculty of Medicine and Pharmacy, University of Oradea, 1 December Sq., 410081 Oradea, Romania; lucidaina@gmail.com; 4Bihor Emergency Clinical County Hospital, 410087 Oradea, Romania

**Keywords:** patient satisfaction, hospital, postoperative care, overall impression

## Abstract

An analysis of patient satisfaction in the context of healthcare reflects the patient’s perception of actual care through the prism of their expectations of ideal care. A study was conducted to investigate patient satisfaction with postoperative care in the context of improvements in hospital medical services (renovations and modernization of operating rooms and wards with beds, endowments in medical equipment for investigation and treatment, new work procedures, and revisions of existing procedures). Statistical analysis was performed based on the questionnaires distributed to patients hospitalized in the surgical wards of the Bihor County Emergency Clinical Hospital, between 2019 and 2023. A total of 4018 questionnaires were collected, and after the elimination of questionnaires with incomplete or incorrect data, 3985 remained in the analysis. Of the total of 2994 people who reported having undergone surgery, a total of 2090 responded to the questions that focused on postoperative care and overall impression of the hospital. No statistically significant differences in patient satisfaction by sociodemographic factors were found. A statistically significant increase in patient satisfaction with postoperative care and overall impression of the hospital was observed over the last 5 years. Correlation testing between postoperative care responses and overall impression, using the Sperman method, shows a directly proportional relationship between the two variables. In the future, it is necessary to extend the patient satisfaction questionnaire to comprehensively analyze the surgical component in order to identify gaps in postoperative care, helping decision makers to improve the medical services provided to patients.

## 1. Introduction

In recent years, the specialized literature has been enriched with numerous studies that bring to the attention of specialists the various analyses on patient satisfaction [1,2,3,4]. The various approaches related to this topic start from two basic considerations: from the patient’s perspective—it reflects the patient’s experience in the hospital—and from the healthcare provider’s perspective—it provides a picture of the organization’s efficiency [5,6].

Although epidemiological studies may not be spectacular, they are very useful in informing medical care directives. Recognizing and managing protocol gaps lead to the improvement in medical services, which in turn increases patient satisfaction and implicitly their trust in the entire medical system.

Patient satisfaction is one of the three dimensions of healthcare quality management, along with professional competence and total quality management, and is a useful tool in the evaluation of health services and patient care [7,8].

At the national and international levels, there is no standard patient satisfaction questionnaire, but only recommendations on its content. In Romania, hospitals are mandated to measure patient satisfaction on an ongoing basis, and the National Authority for Quality Management in Healthcare indicates the standards that the questionnaire should maintain. In addition to these recommendations, each hospital or healthcare facility is free to complete the questionnaire according to its needs [9,10,11].

In the Bihor County Emergency Clinical Hospital (SCJUBh), there is a continuous monitoring of patient satisfaction and a continuous adaptation of the questionnaire in order to capture the most important aspects that lead to improving the quality of medical services. Such analyses reflect the results obtained and the measures taken.

The recent infrastructural and procedural changes in Romania’s healthcare system, driven by critical challenges such as insufficient funding, staff shortages, and service delivery gaps, have aimed to modernize facilities and streamline healthcare processes. These changes are vital to ensuring equitable and accessible healthcare, addressing the growing needs of the population, and improving the efficiency and quality of care across both public and private healthcare sectors [12].

Due to the complex investments that have taken place in the hospital (refurbishment of spaces, equipping with high-performance equipment, etc.) correlated with the implementation of a series of work procedures, it was necessary to conduct a series of analyses to verify the extent to which these changes are reflected in patient care. Such an analysis can also be found in the present study, which is focused on the activity in the surgical wards and the operating room.

SCJUBh is a tertiary level hospital in the municipality of Oradea, located in Northwestern Romania. The hospital has 1865 beds, being one of the largest hospitals in Romania and providing medical services in continuous hospitalization, day hospitalization, and outpatient care, as well as providing medical and surgical services in emergencies to the Bihor County.

In Romania, hospitals are mandated to measure patient satisfaction regularly, following guidelines set by the National Authority for Quality Management in Healthcare [13]. However, unlike many other countries where standardized patient satisfaction tools such as the Hospital Consumer Assessment of Healthcare Providers and Systems (HCAHPS) are used, Romanian hospitals have flexibility in designing their questionnaires [14]. This flexibility contrasts with the more rigid standards seen in countries like the US or the UK [15].

The first major investments started 10 years ago, when a European-funded project started renovations and modernizations in the operating rooms, and 5 years ago, energy efficiency works, rehabilitation, and modernization of room quality started. The period of the COVID pandemic led to the administering of medical care only in cases of medical–surgical emergencies, the creation of new functional circuits intended for patients infected with SARS-CoV-2, and stagnation in the revision of existing protocols and working procedures in the hospital [16,17]. After March 2022, the current activities gradually resumed in the hospital, with the number of discharged patients being similar to that recorded in 2019.

Given the particularities described above, the analysis of patient satisfaction in the context of healthcare reflects the patient’s perception of the actual care through the prism of the patient’s expectations of ideal care [18,19]. This can provide a relevant assessment of the healthcare provided at a hospital [20]. Also, by analyzing patient satisfaction, the patient becomes an active actor in the healthcare system, participating in medical decisions related to investigations and treatment [21].

The importance given to this area is reflected in the multiple published studies, with patient satisfaction varying according to the variables analyzed. For example, for operated patients, a satisfaction level of 95% was recorded in Portugal [22], 91% in Germany [23], 86.7% in Iran [24], 86.6% in the UK [25], 78.8% in Romania [10,26], 77% in Spain [27], and 75% in Australia [28]. The results obtained in published studies are diverse, without being able to establish a relationship between patients’ experience and the quality of clinical care [29,30]. In order to reflect as accurately as possible the quality of care provided to patients, it is recommended that each hospital dynamically evaluates patient satisfaction questionnaires. It is also very important that the questions in the content of the questionnaire are simple, clear, and concise, and that the interview administrator provides support to a patient when needed. Another important aspect is the regular completion, verification, and validation of the patient satisfaction questionnaire.

While numerous studies have assessed patient satisfaction in various healthcare settings, few have focused on postoperative care in a large tertiary hospital in Romania, especially in light of recent infrastructural and procedural changes. This study aims to fill that gap by providing a detailed analysis of patient satisfaction and its evolution over a five-year period, incorporating feedback on both surgical care and overall hospital experience. Such an analysis was carried out in the present study, targeting patient satisfaction with postoperative care, in the context of the improvement in medical services in the hospital (renovations and modernization of operating rooms and wards with beds, endowments in medical equipment for investigation and treatment, and new working procedures and revisions of existing procedures). The development of an algorithm for patient satisfaction provides a straightforward way to evaluate differences between departments or hospitals. Simplifying the calculations with the help of a complex algorithm allows for fast and efficient statistical processing.

## 2. Materials and Methods

### 2.1. Study Design

This study utilized a retrospective observational design, analyzing patient satisfaction data collected from questionnaires distributed to patients hospitalized in the surgical wards of Stationary I at Bihor County Emergency Hospital between 2019 and 2023. A total of 4018 questionnaires were collected, and after eliminating the questionnaires with incomplete or incorrect data, 3985 remained in the analysis. Incomplete or incorrect questionnaires were identified through automated checks for missing data and logical inconsistencies in responses. These questionnaires were subsequently excluded from the analysis to ensure data quality and accuracy. The questionnaires with over 20% missing responses were discarded.

The patient satisfaction questionnaire was initially developed based on the existing models used in Romanian hospitals, with adaptations made to address specific aspects of postoperative care. The questionnaire was validated through expert reviews and pilot testing with a small group of patients. Cronbach’s alpha was calculated to assess internal consistency. For non-Romanian-speaking patients, the questionnaire was translated using the forward and backward translation method to ensure accuracy and consistency.

Responses to 3 questions were tracked. The first question (“During your hospitalization did you undergo surgery?”) is a dichotomic question, and the second (“How do you rate the postoperative care and medical care provided in the Intensive Care Unit (if any)?”) and the third question (“Your overall impression of the hospital?”) are structured on a Likert scale from 1 to 5.

Modifications to the patient satisfaction questionnaire were based on feedback from healthcare professionals and patient responses, aiming to capture evolving aspects of patient care. The questionnaire underwent iterative revisions, followed by pilot testing to ensure clarity and relevance. Validation involved expert reviews and internal consistency checks (e.g., Cronbach’s alpha) to ensure the reliability of the questions.

### 2.2. Algorithm of Patient Satisfaction

The specific weights for gender, age, and environmental factors were developed based on internal hospital data and expert input. While no existing studies provide exact values for these factors in the context of Romanian hospitals, these weights reflect observed patterns in patient recovery times and satisfaction across demographic groups. Although there is no specific algorithm that we used, the specialized literature provides data showing significantly improved results in developing an algorithm tailored to the specific area [31,32].

To improve patient satisfaction, it is essential to identify gaps in postoperative care. For example, feedback regarding postoperative pain management resulted in the implementation of new pain control protocols, leading to higher patient satisfaction in subsequent assessments. Therefore, we developed a scoring system based on municipal emergency hospital statistics. The score comprises three parts:

1.Demographic Factors:

The gender, age, and environmental scores were adapted from studies that highlight recovery rate differences based on these factors [33].

Gender score: male = 1.2 and female = 1.0, based on the existing statistics indicating that women recover faster.

Age score: 1.0 + (0.1 ∗ (patient age/10)), where older age corresponds to slower recovery.

Environmental score: urban = 1.3 and rural = 1.0.

These scores were developed based on internal hospital data and the relevant literature indicating that gender, age, and environmental factors influence recovery and satisfaction.

2.Medical Factors:

Infection score: with infections = 1.5 and without infections = 1.0.

Complication score: with complications = 2.0 and without complications = 1.0.

Protocol score: standard = 1.0 and personalized/innovative protocol = 0.8.

Innovation score: no innovative products = 1.0 and innovations = 0.9.

3.Pain Factor:

Pain score: 1.0 + (0.1 × patient’s pain score), assuming the pain score is on a scale of 1 to 10 according to the international standard.

Gender was included in the scoring system as studies indicate differences in recovery times and satisfaction levels between men and women. Age was considered due to its known influence on recovery rates, with older patients typically experiencing slower recoveries. The environmental factor (urban vs. rural) was included to account for differences in healthcare access and expectations between these populations.

Total Score Calculation:Algorithm_Total score_ = (Gender score × Age score × Environmental score × Infection score × Complication score × Protocol score × Innovation score × Pain score)(1)

Interpretation of the score:

Low satisfaction: total score > 3.0;

Moderate satisfaction: total score > 1.5 and ≤3.0;

High satisfaction: total score ≤ 1.5.

The COVID-19 pandemic had a profound impact on healthcare delivery and patient satisfaction due to restricted access to hospitals, delays in non-emergency surgeries, and altered protocols. Patient feedback during this period highlighted concerns about longer wait times and anxiety related to the pandemic, which influenced the overall satisfaction scores.

### 2.3. Statistical Analysis

Chi-square, Fisher’s exact test, ANOVA, and Tukey’s HSD test were used to determine the statistical significance of the results. Tukey’s HSD test was applied post hoc to identify specific differences between the means of multiple groups, particularly in patient satisfaction scores across different wards. The Spearman correlation coefficient was used to determine the level of dependence between the answers to questions 2 and 3. The confidence interval was set at 95%. Statistical analysis was performed using the program R version 4.3.1 [34].

Patient satisfaction data were analyzed using a combination of descriptive and inferential statistics. The chi-square test was used to analyze categorical variables, while ANOVA and Tukey’s HSD test were employed to compare mean satisfaction scores between different wards and time periods. Spearman’s correlation was used to measure the relationship between patients’ ratings of postoperative care and the overall hospital impression. A custom algorithm was developed to calculate satisfaction scores, factoring in variables such as gender, age, and environmental factors.

### 2.4. Participants

Patients were selected using convenience sampling, with at least 10% of those hospitalized in the surgical wards during the study period receiving the questionnaire, and the sample size was determined based on previous studies on patient satisfaction in similar settings, ensuring adequate power and representativeness across different wards and demographics. To ensure representativeness, questionnaires were distributed evenly across different wards and patient demographics.

In the SCJUBh, a number of satisfaction questionnaires are distributed monthly within each ward and compartment to at least 10% of the hospitalized patients. Thus, 3985 patient satisfaction questionnaires distributed in the period 2019–2023 were analyzed at the wards with a surgical profile within Stationary I, distributed by year; see Table 1.

Surgical wards with ≥25 beds were selected: plastic surgery, reconstructive microsurgery, general surgery (2 wards), neurosurgery, otolaryngology (ENT), orthopedics and traumatology (2 wards), and urology.

The age of respondents ranged from 12 to 96 years with a mean of 53.03 years and a standard deviation of 15.8 years. The gender distribution of the questionnaires was in favor of males who responded to 53.15% of the questionnaires. Overall, 38.87% of the respondents were from rural areas and 44.47% from urban areas, and the rest did not fill in the answer to this question. The distribution of the respondents by educational level is provided in Table 2.

## 3. Results

The distribution of questionnaires by year and ward of origin is described in Table 3. Most questionnaires were distributed in wards with 35 beds and more: general surgery I (16.91%), urology (13.63%), and general surgery II (12.6%).

The mean age of the respondents, described in Table 4, varies according to the profile of the surgical ward. The differences are statistically significant (ANOVA test, *p* < 0.00001).

Between 40% and 99% of those surveyed said that they had been operated on by answering “yes” to the first of the analyzed questions (Table 5).

Out of the total of 2994 persons who stated that they had undergone surgery, 2090 answered the second analyzed question. The questions assessing the postoperative care and the general impression of the Hospital, being structured on a Likert scale from 1 to 5, were analyzed. The mean scores of the responses by ward and year are shown in Table 6.

The test of statistical significance for the differences obtained at each ward and cumulatively between years in the analyzed period was performed using the ANOVA test, and the results are presented in Table 6.

The appreciation of postoperative care was not influenced by the patient’s sex (4.78 for women and 4.79 for men) or residence (4.80 for rural and 4.81 for urban). The level of schooling did not influence the appreciation of postoperative care, with the following values: primary education—4.68; secondary education—4.77; high school—4.82; and university—4.82. Even if there were small differences, they were not statistically significant (ANOVA test, *p* > 0.05).

### 3.1. Evaluation of Satisfaction

The postoperative period is a critical phase during which patients are vulnerable to complications and significant psychological stress. The quality of care provided during this period can directly influence patients’ perception of the entire surgical process. Factors such as pain management, infection prevention, effective communication, and emotional support contribute to overall patient satisfaction. Studies show that patients who receive high-quality postoperative care report increased satisfaction and a better quality of life after surgery.

Patients were evaluated using the algorithm. Between 2019 and 2023 (Figure 1), variations were observed in the mean values and standard deviations of a given measurement. The blue box represents the interquartile range (IQR) of the patient satisfaction scores, while the red line indicates the median score. In 2019, the mean was 2.09 with a standard deviation of 0.52, suggesting moderate variability in individual values from the overall mean. In 2020, the mean increased slightly to 2.16 with a standard deviation of 0.55, indicating a slight increase in data dispersion.

In 2021, there was a notable drop in the mean to 1.70, accompanied by a significant reduction in the standard deviation to 0.18, showing a higher concentration of values around the mean. In 2022, the mean increased slightly to 1.78 with a standard deviation of 0.42, signaling an increase in variability compared to the previous year. In 2023, the mean was the lowest of the analyzed period at 1.50, and the standard deviation was 0.14, indicating very little variability in the data collected.

Analysis of the data in Table 7, “Paired Samples Test”, reveals mean differences and significant correlations between the years studied, providing an overview of the evolution of the measurements over time. Notably, the differences between the 2019 and 2023 values were remarkable. The mean difference of 0.58667, associated with a standard deviation of 0.38857, was supported by a 95% confidence interval with bounds between 0.17889 and 0.99445. This indicates a significant positive change, confirmed by a *t*-test that revealed a significant value (t = 3.698, *p* = 0.014). This difference highlights a notable change in the measured values, underscored by clear statistical significance.

Another notable finding in our analysis was the difference between the years 2020 and 2023. The observed mean difference was 0.65167, with a standard deviation of 0.51316. The 95% confidence interval for this difference ranged from 0.11313 to 1.19020, suggesting a significant change between these two years. The t-test confirmed this observation, indicating a significant value (t = 3.111, *p* = 0.027), reinforcing the idea that there was considerable variation in the data collected between 2020 and 2023.

On the other hand, other comparisons, such as that between the years 2019 and 2020, showed an insignificant mean difference of −0.06500, with a t-test that did not indicate statistical significance (t = −0.259, *p* = 0.806). Similarly, although a mean difference of 0.39167 was observed between 2019 and 2021, it did not reach the threshold of statistical significance (t = 1.541, *p* = 0.184). Likewise, the mean difference between 2020 and 2021 of 0.45667 was close to being statistically significant but did not meet the criteria (t = 2.276, *p* = 0.072).

### 3.2. Correlations

The distribution of responses to questions 2 and 3 was tested for normality using the Shapiro–Wilk test, which indicated a lack of normal distribution (*p* < 0.05). Therefore, we applied Spearman’s correlation, a non-parametric test suitable for non-normally distributed data. The moderate positive correlation (r = 0.51) between questions 2 and 3 suggests that patients who rated postoperative care highly also tended to rate their overall impression of the hospital favorably, indicating some interdependence between the two questions. This correlation suggests that the two questions are related, measuring different but complementary aspects of patient satisfaction. However, this does not directly assess the clarity or construction quality of the questions. The graphical representation confirms this (Figure 2).

## 4. Discussion

Measuring patient satisfaction with postoperative care helps to identify dysfunctions in patient care. The results obtained are an important source of information for hospital management in making decisions to improve patient satisfaction and their overall impression of the hospital environment.

The published research in this area is diverse, with each analysis of patient satisfaction focusing on the particularities considered important for each hospital. Therefore, as there is no single model for analyzing patient satisfaction, there is still a lack of evidence related to patients who have undergone surgery. Various rating scales are used in patient satisfaction [35]. Without further explanation by the interviewer, it is difficult for patients to evaluate the technical quality of care [36,37]. Therefore, measuring the patient’s postoperative care experience is more complex than identifying an existing isolated question in the patient satisfaction questionnaire [38].

In this study, we assessed patient satisfaction with postoperative care that includes data on patient information, staff–patient relationship, discomfort and needs, and fear and worry. These items were explained by the interviewer when the questionnaire was administered.

There was a statistically significant difference in the mean age of the respondents according to the profile of the surgical department. The lowest mean age is found in plastic surgery (47.15 years) and otorhinolaryngology (48.30 years) wards, while the highest mean age corresponds to orthopedics and traumatology I (57.65 years) and II (56.89 years) wards, respectively. This fact is explained by the specific pathology of each department, especially accidents in young people (plastic surgery), frequent pediatric pathology in children (otorhinolaryngology), and degenerative pathology in the elderly (orthopedics and traumatology).

Postoperative care is rated favorably by patients between 2019 and 2022, with scores ranging from 4.23 (general surgery I, 2019) to 5 (otolaryngology, 2021). Patient satisfaction with postoperative care is increased in multiple published studies [1,10,22,23,24,25,26,27,39], which can be explained by the considerable improvement in postoperative patient’s health status compared to their previous state [1,2].

When analyzed dynamically, postoperative care has an upward trend in the analyzed surgical wards, with the exception of orthopedics and traumatology II. Differences between specialties are also described in other similar studies. A study published in 2008 shows significant differences between specialties, with satisfaction scores ranging from 82.5 in plastic surgery to 99.00 in urology [40].

The performed analysis found no significant association between sociodemographic data such as gender, age, residence, educational status, and patient satisfaction with postoperative surgical ward care, similar to other studies [41]. The literature data on this issue are diverse, and different studies do not support a significant association, while other studies emphasize the significant association between sociodemographic variables and patient satisfaction [18,21,42,43]. Although the results differ in the importance of demographic and social demographic factors in determining patient satisfaction, these analyses may provide important insights for the hospital on how certain demographic factors may influence patient satisfaction.

We find studies in the literature that have identified the significant influence of gender and age on patient satisfaction [44,45,46]. In these studies, the results also differ; thus, we find that, on the one hand, men are generally more satisfied than women with the care provided [1,33], which contrasts with other studies that found that women are more satisfied than men with the care provided in surgical wards [4,21]. Satisfaction in patients over 50 years of age was higher than that in younger patients [40,44].

In our study, no statistically significant differences in satisfaction with care based on patient’s residence and educational status were observed. The obtained data contrast with those of other studies in which rural patients are more satisfied [32] or studies showing higher satisfaction of urban patients [1]. Patient satisfaction decreases as the educational level increases [1].

The study found that postoperative patient satisfaction directly correlates with the overall impression of the hospital. The results indicated an overall high level of satisfaction corresponding to a high overall impression of the hospital, which was also consistent with the level of postoperative patient satisfaction. The results regarding the overall impression are similar to those of other previously conducted studies [1,3,4,47,48,49,50]. The high level of postoperative patient satisfaction may be explained by the hopeful attitude of patients recovering postoperatively [1]. These changes indicate considerable variation in the collected data, suggesting the possible influence of specific factors that distinctly affected the measurements during these years. This development underscores the importance of continuous monitoring and rigorous analysis to understand and manage data variability in complex contexts.

Although patient satisfaction with postoperative care and the overall impression of the hospital has shown a statistically significant upward trend, correlated with measures to improve the provided medical services, the analysis does not provide a comprehensive understanding of the entire surgical care experience. In the future, the patient satisfaction questionnaire should be expanded to include a comprehensive analysis of the surgical component: preoperative preparation, pre-anesthetic consultation, the surgical procedure itself, and the post-surgery period. Surveys would provide more comprehensive information on the factors impacting patient satisfaction with surgical care, helping decision makers to improve the healthcare provided to patients.

## 5. Conclusions

Patient satisfaction with postoperative care is a useful tool to evaluate the medical services provided in the hospital. In the context of the significant investments made in recent years in operating theaters and wards with beds, patient assessment is the most important measure providing feedback on these investments. Over the last 5 years, a statistically significant increase in patient satisfaction with postoperative care and the overall impression of the hospital has been observed. No statistically significant differences in patient satisfaction by sociodemographic factors were found. In order to identify gaps in postoperative care, the patient satisfaction questionnaire should be expanded in the future, and further research in this area should be conducted. The most significant differences occurred between 2019 and 2023, as well as between 2020 and 2023, according to the satisfaction algorithm. This indicates a substantial increase in patient satisfaction.

## Figures and Tables

**Figure 1 healthcare-12-01824-f001:**
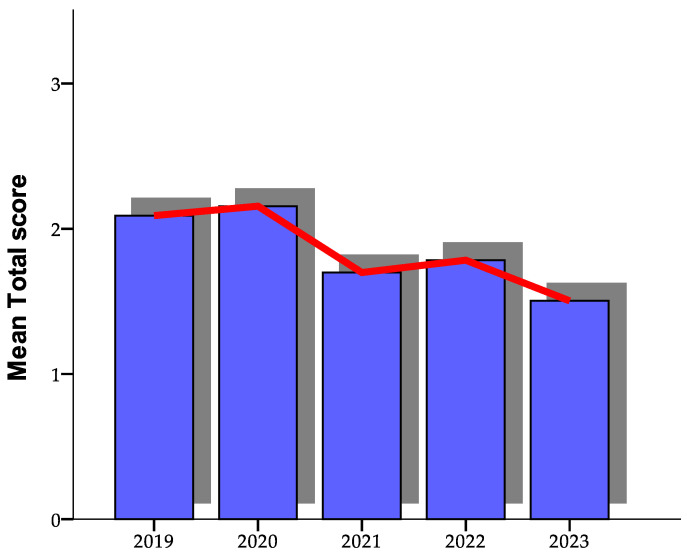
Evolution of the total patient satisfaction score over the 5-year study period.

**Figure 2 healthcare-12-01824-f002:**
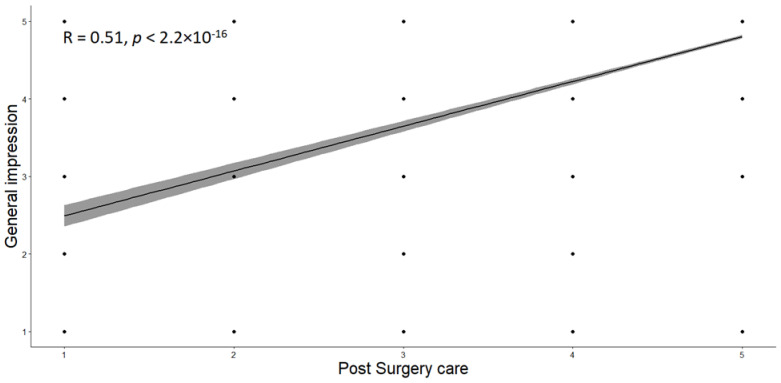
Correlation between appreciation of postoperative care and overall impression.

**Table 1 healthcare-12-01824-t001:** Annual distribution of analyzed questionnaires.

Year	2019	2020	2021	2022	2023	Total
Number of questionnaires	818	858	731	553	1025	3985

**Table 2 healthcare-12-01824-t002:** Distribution of completed questionnaires by level of schooling.

Education Level	Primary	Secondary	High School	University	No Answer
% completed questionnaires	5.20%	15.63%	52.20%	23.41%	3.56%

**Table 3 healthcare-12-01824-t003:** Annual distribution of questionnaires by ward.

Section/Year	2019	2020	2021	2022	2023	Total
Clinical Department of Plastic Surgery, Reconstructive Microsurgery	112	143	95	68	78	496
Clinical Department of General Surgery I	126	156	133	97	162	674
Clinical Department of General Surgery II	112	129	83	57	121	502
Clinical Department of Neurosurgery	94	111	81	58	81	425
ENT Clinical Department	85	82	84	57	173	481
Clinical Department of Orthopedics and Trauma I	88	67	82	72	151	460
Clinical Department of Orthopedics and Trauma II	96	73	56	54	125	404
Clinical Department of Urology	105	97	117	90	134	543
Total	818	858	731	553	1025	3985

**Table 4 healthcare-12-01824-t004:** Average age in years according to surgical department profile.

Section/Year	2019	2020	2021	2022	2023	Total
Clinical Department of Plastic Surgery, Reconstructive Microsurgery	43.05	46.92	48.33	53.22	46.73	47.15
Clinical Department of General Surgery I	53.48	52.67	52.50	49.87	49.13	51.55
Clinical Department of General Surgery II	56.40	52.25	55.53	51.35	54.64	54.19
Clinical Department of Neurosurgery	55.20	53.90	53.78	55.09	52.91	54.14
ENT Clinical Department	49.45	46.96	51.71	45.33	47.68	48.30
Clinical Department of Orthopedics and Trauma I	57.52	59.69	54.24	57.79	58.66	57.65
Clinical Department of Orthopedics and Trauma II	56.24	60.74	54.57	56.94	56.07	56.89
Clinical Department of Urology	57.02	56.61	56.49	50.89	57.44	55.88

**Table 5 healthcare-12-01824-t005:** Percentage of people who had surgery.

Section/Year	2019	2020	2021	2022	2023	Total
Clinical Department of Plastic Surgery, Reconstructive Microsurgery	99.11%	99.30%	96.84%	88.24%	88.46%	95.56%
Clinical Department of General Surgery I	63.49%	66.03%	61.65%	81.44%	82.10%	70.77%
Clinical Department of General Surgery II	80.36%	75.97%	83.13%	87.72%	78.51%	80.08%
Clinical Department of Neurosurgery	45.74%	47.75%	54.32%	46.55%	53.09%	49.41%
ENT Clinical Department	52.94%	40.24%	57.14%	50.88%	61.27%	54.26%
Clinical Department of Orthopedics and Trauma I	90.91%	88.06%	87.80%	91.67%	90.73%	90.00%
Clinical Department of Orthopedics and Trauma II	81.25%	90.41%	76.79%	94.44%	92.80%	87.62%
Clinical Department of Urology	72.38%	76.29%	76.92%	76.67%	69.40%	74.03%
Total	73.72%	73.19%	73.87%	77.94%	77.27%	75.13%

**Table 6 healthcare-12-01824-t006:** Average answers to questions 2 and 3.

Section		2019	2020	2021	2022	2023	ANOVA Test, *p*-Value
Clinical Department of Plastic Surgery, Reconstructive Microsurgery	postoperative care	4.93	4.98	4.99	4.88	4.97	0.54
	overall impression	4.92	4.88	4.84	4.88	4.78	<0.05
Clinical Department of General Surgery I	postoperative care	4.23	4.69	4.78	4.83	4.8	<0.001
	overall impression	4.11	4.69	4.56	4.6	4.74	<0.01
Clinical Department of General Surgery II	postoperative care	4.93	4.78	4.82	4.88	4.99	0.22
	overall impression	4.87	4.73	4.94	4.85	4.93	0.08
Clinical Department of Neurosurgery	postoperative care	4.71	4.83	4.64	4.82	4.75	0.91
	overall impression	4.59	4.59	4.39	4.65	4.7	0.63
ENT Clinical Department	postoperative care	4.71	4.96	5	4.78	4.96	0.24
	overall impression	4.86	4.74	4.95	4.7	4.92	0.25
Clinical Department of Orthopedics and Trauma I	postoperative care	4.7	4.63	4.84	4.98	4.79	0.13
	overall impression	4.6	4.39	4.65	4.79	4.78	<0.01
Clinical Department of Orthopedics and Trauma II	postoperative care	4.63	4.42	4.82	4.61	4.35	0.14
	overall impression	4.37	4.15	4.68	4.54	4.14	0.36
Clinical Department of Urology	postoperative care	4.78	4.8	4.76	4.71	4.85	0.76
	overall impression	4.62	4.69	4.65	4.53	4.79	0.36
Cumulative	postoperative care	4.71	4.78	4.84	4.81	4.8	<0.05
	overall impression	4.62	4.66	4.72	4.67	4.73	<0.05

**Table 7 healthcare-12-01824-t007:** Paired samples test for the 5-year study period.

Paired Samples Test		Paired Differences				t	*p*
		Mean	SD	95% CI			
				Lower	Upper		
Pair 1	2019–2020	−0.06500	0.61452	−0.70990	0.57990	−0.259	0.806
Pair 2	2019–2021	0.39167	0.62252	−0.26163	1.04497	1.541	0.184
Pair 3	2019–2022	0.30667	0.62073	−0.34475	0.95808	1.210	0.280
Pair 4	2019–2023	0.58667	0.38857	0.17889	0.99445	3.698	0.014 *
Pair 5	2020–2021	0.45667	0.49156	−0.05919	0.97252	2.276	0.072
Pair 6	2020–2022	0.37167	0.66508	−0.32630	1.06963	1.369	0.229
Pair 7	2020–2023	0.65167	0.51316	0.11313	1.19020	3.111	0.027 *
Pair 8	2021–2022	−0.08500	0.52199	−0.63279	0.46279	−0.399	0.706
Pair 9	2021–2023	0.19500	0.26067	−0.07856	0.46856	1.832	0.126
Pair 10	2022–2023	0.28000	0.45060	−0.19288	0.75288	1.522	0.188

SD = standard deviation; CI = confident interval; t = coefficient of t-Student test; *p* = statistical significance, * = Correlation is significant at the 0.05 level (2-tailed).

## Data Availability

Data are contained within the article.

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
