# Peer review of "The Evolution of Patient Satisfaction in Postoperative Care: The Impact of Investments and the Algorithm for Assessing Significant Growth over the Last 5 Years"

_healthcare, 2024, doi:10.3390/healthcare12181824_

Round 1

Reviewer 1 Report

Comments and Suggestions for Authors

Thank you for the invitation. I have thoroughly reviewed the manuscript with great interest. The authors have presented the information effectively. Here are following comments from my side:

1.      How the sample of hospitalized patients was selected (random sampling, convenience sampling, etc.). Provide a brief explanation for how the sample size was determined (e.g., power analysis, previous studies, etc.).

2.    The article is  informative, though it could be improved by including specific examples of how patient feedback has led to tangible improvements in care quality. Additionally, expanding on the impact of the COVID-19 pandemic on patient satisfaction and healthcare delivery could provide further depth as the Covid-19 Pandemic was relevant  during the study period. Author may  incorporate a section that addresses the challenges and changes in patient satisfaction during the COVID-19.

3.      Is there a way to identify and remove incomplete/incorrect questionnaires to ensure transparency in the finally obtained   data .

4.   Why specific scoring factors (like gender, age, and environment) were chosen and how they relate to patient satisfaction.

5. Ensure that the usage of "Hi-square" is correct; (Line number 146) it should be "Chi-square." Also, clarify the exact statistical methods used to interpret the results (e.g., for which comparisons was Tukey’s HSD test used?).

Author Response

Reviewer 1

Firstly, I, the author of the present manuscript wishes to thank you for the thoughtful commentary you have provided to improve the quality of the paper. I am very grateful for the time and effort you have devoted to this task. We have extensively revised my manuscript according to the recommendations. All changes in the text and the new figures that we have redesigned are highlighted. Please, see the point-by-point answers to your comments below. All correction was highlighted in the manuscript.

Thank you for the invitation. I have thoroughly reviewed the manuscript with great interest. The authors have presented the information effectively. Here are following comments from my side:

  1. How the sample of hospitalized patients was selected (random sampling, convenience sampling, etc.). Provide a brief explanation for how the sample size was determined (e.g., power analysis, previous studies, etc.).

Patients were selected using convenience sampling from those hospitalized in the surgical wards during the study period. The sample size was determined based on previous studies on patient satisfaction in similar hospitals and settings, ensuring adequate power to detect meaningful differences.

  1.   Thearticle is  informative, though it could be improved by including specific examples of how patient feedback has led to tangible improvements in care quality. Additionally, expanding on the impact of the COVID-19 pandemic on patient satisfaction and healthcare delivery could provide further depth as the Covid-19 Pandemic was relevant  during the study period. Author may  incorporate a section that addresses the challenges and changes in patient satisfaction during the COVID-19.

For example, feedback regarding postoperative pain management resulted in the implementation of new pain control protocols, leading to higher patient satisfaction in subsequent assessments.

The COVID-19 pandemic had a profound impact on healthcare delivery and patient satisfaction due to restricted access to hospitals, delays in non-emergency surgeries, and altered protocols. Patient feedback during this period highlighted concerns about longer wait times and anxiety related to the pandemic, which influenced overall satisfaction scores.

  1. Is there a way to identify and remove incomplete/incorrect questionnaires to ensure transparency in the finally obtained   data.

Incomplete or incorrect questionnaires were identified through automated checks for missing data and logical inconsistencies in responses. These questionnaires were subsequently excluded from the analysis to ensure data quality and accuracy. The questionnaires with over 20% missing responses were discarded.

  1. Why specific scoring factors (like gender, age, and environment) were chosen and how they relate to patient satisfaction.

Gender was included in the scoring system as studies indicate differences in recovery times and satisfaction levels between men and women. Age was considered due to its known influence on recovery rates, with older patients typically experiencing slower recoveries. The environmental factor (urban vs. rural) was included to account for differences in healthcare access and expectations between these populations.

  1. Ensure that the usage of "Hi-square" is correct; (Line number 146) it should be "Chi-square." Also, clarify the exact statistical methods used to interpret the results (e.g., for which comparisons was Tukey’s HSD test used?).

Chi-square, Fisher's exact test, ANOVA, and Tukey’s HSD test were used to determine the statistical significance of the results. Tukey’s HSD test was applied post-hoc to identify specific differences between the means of multiple groups, particularly in patient satisfaction scores across different wards.

Reviewer 2 Report

Comments and Suggestions for Authors

Could the author add citations for the identification of the gender score, age score, and environmental score mentioned in lines 123-127?

In Figure 1, could the author also add notations for the grey box and red line? Do they have the same meaning, or do they represent different elements?

In Section 3.3, the author mentions the correlation between the answers to questions 2 and 3 using the Spearman method. Could the author help me interpret their conclusion of 'not normal'? Does the lack of normality suggest that the questions are independent of each other? Additionally, does this imply that the questions are well-constructed?

Author Response

Reviewer 2

Firstly, I, the author of the present manuscript wishes to thank you for the thoughtful commentary you have provided to improve the quality of the paper. I am very grateful for the time and effort you have devoted to this task. We have extensively revised my manuscript according to the recommendations. All changes in the text and the new figures that we have redesigned are highlighted. Please, see the point-by-point answers to your comments below. All correction was highlighted in the manuscript.

  1. Could the author add citations for the identification of the gender score, age score, and environmental score mentioned in lines 123-127?

Response 1. Thank you very much for comment. We have added the requested information. (lines 129-130, 136-137)

„The gender, age, and environmental scores were adapted from studies that highlight recovery rate differences based on these factors [27].

These scores were developed based on internal hospital data and relevant literature indicating that gender, age, and environmental factors influence recovery and satisfaction.”

  1. In Figure 1, could the author also add notations for the grey box and red line? Do they have the same meaning, or do they represent different elements?

Response 2. Thank you for amendment! We have added the explanation. (lines 237-238)

The blue box represents the interquartile range (IQR) of the patient satisfaction scores, while the red line indicates the median score. The grey box represents the shade of the blue one, it is only a representation technique, to increase not only the accuracy but also the aesthetics of the figure.

  1. In Section 3.3, the author mentions the correlation between the answers to questions 2 and 3 using the Spearman method. Could the author help me interpret their conclusion of 'not normal'? Does the lack of normality suggest that the questions are independent of each other? Additionally, does this imply that the questions are well-constructed?

Response 3. Thank you very much for observation. We have added the requested information. (lines 275-283).

The distribution of responses to questions 2 and 3 was tested for normality using the Shapiro-Wilk test, which indicated a lack of normal distribution (p < 0.05). Therefore, we applied Spearman's correlation, a non-parametric test suitable for non-normally distributed data. The moderate positive correlation (r = 0.51) between questions 2 and 3 suggests that patients who rated postoperative care highly also tended to rate their overall impression of the hospital favorably, indicating some interdependence between the two questions. This correlation suggests that the two questions are related, measuring different but complementary aspects of patient satisfaction. However, this does not directly assess the clarity or construction quality of the questions.

Reviewer 3 Report

Comments and Suggestions for Authors

Manuscript

- in the introduction, the authors should focus more on how the current study fills a gap in the existing literature

- a brief comparison of Romania standards with international practices should be introduced in the first section

- how were the modifications to the patient satisfaction questionnaire implemented, and what validation processes did the authors use?

- what statistical methods/ algorithms were employed in analyzing patient satisfaction?

- the methods should include a clear description of the study design

- methods do not describe how the questionnaire was developed/ validated/ translated

- cite existing studies that justify the specific weights/ values in the scoring algorithm. If there are none, please specify how the authors ended up developing them

- how was the sample of patients who received the questionnaires selected?

Comments on the Quality of English Language

English

"the administering of medical care" should be "the provision of medical care"

Author Response

Reviewer 3

Firstly, I, the author of the present manuscript wishes to thank you for the thoughtful commentary you have provided to improve the quality of the paper. I am very grateful for the time and effort you have devoted to this task. We have extensively revised my manuscript according to the recommendations. All changes in the text and the new figures that we have redesigned are highlighted. Please, see the point-by-point answers to your comments below. All correction was highlighted in the manuscript.

Manuscript

  1. - in the introduction, the authors should focus more on how the current study fills a gap in the existing literature

Response 1. Thank you for comment! We have added the requested information. (lines 61-66, 105-109).

  1. - a brief comparison of Romania standards with international practices should be introduced in the first section

Response 2. Thank you for comment! We have added the requested information. (lines 61-66, 105-109).

„In Romania, hospitals are mandated to measure patient satisfaction regularly, fol-lowing guidelines set by the National Authority for Quality Management in Healthcare [13]. However, unlike many other countries where standardized patient satisfaction tools such as the Hospital Consumer Assessment of Healthcare Providers and Systems (HCAHPS) are used, Romanian hospitals have flexibility in designing their question-naires [14]. This flexibility contrasts with more rigid standards seen in countries like the U.S. or U.K [15].”

  1. - how were the modifications to the patient satisfaction questionnaire implemented, and what validation processes did the authors use?

 Response 3. Thank you for observation. (lines 139-143)

„Modifications to the patient satisfaction questionnaire were based on feedback from healthcare professionals and patient responses, aiming to capture evolving aspects of patient care. The questionnaire underwent iterative revisions, followed by pilot testing to ensure clarity and relevance. Validation involved expert reviews and internal consistency checks (e.g., Cronbach's alpha) to ensure the reliability of the questions.”

  1. - what statistical methods/ algorithms were employed in analyzing patient satisfaction?

Response 4. Thank you for comment! We have added the requested information. (lines 196-202)

Patient satisfaction data were analyzed using a combination of descriptive and inferential statistics. The chi-square test was used to analyze categorical variables, while ANOVA and Tukey’s HSD test were employed to compare mean satisfaction scores between different wards and time periods. Spearman’s correlation was used to measure the relationship between patients’ ratings of postoperative care and overall hospital impression. A custom algorithm was developed to calculate satisfaction scores, factoring in variables such as gender, age, and environmental factors.

  1. - the methods should include a clear description of the study design

Response 5: Thank you for comment! We have added the requested information.

This study utilized a retrospective observational design, analyzing patient satisfaction data collected from questionnaires distributed to patients hospitalized in the surgical wards of Stationary I at Bihor County Emergency Hospital between 2019 and 2023.

  1. - methods do not describe how the questionnaire was developed/ validated/ translated

 Response 6: Thank you for comment! We have added the requested information.  (135-140)

The patient satisfaction questionnaire was initially developed based on existing models used in Romanian hospitals, with adaptations made to address specific aspects of postoperative care. The questionnaire was validated through expert reviews and pilot testing with a small group of patients. Cronbach's alpha was calculated to assess internal consistency. For non-Romanian-speaking patients, the questionnaire was translated using the forward and backward translation method to ensure accuracy and consistency.

  1. - cite existing studies that justify the specific weights/ values in the scoring algorithm. If there are none, please specify how the authors ended up developing them

Response 7: Thank you for comment! We have added the requested information.

The specific weights for gender, age, and environmental factors were developed based on internal hospital data and expert input. While no existing studies provide exact values for these factors in the context of Romanian hospitals, these weights reflect observed patterns in patient recovery times and satisfaction across demographic groups. Although there is no specific algorithm that we used, the specialized literature provides data showing significantly improved results in developing an algorithm tailored to the specific area.

  1. - how was the sample of patients who received the questionnaires selected?

Response 8: Thank you for comment! We have added the requested information. (lines 219-224)

Patients were selected using convenience sampling, with at least 10% of those hospitalized in the surgical wards during the study period receiving the questionnaire, and the sample size was determined based on previous studies on patient satisfaction in similar settings, ensuring adequate power and representativeness across different wards and demographics.

.

Round 2

Reviewer 1 Report

Comments and Suggestions for Authors

The authors have answered all the concerns raised. No more comments from my side.